# Combination Screening of a Naïve Antibody Library Using *E. coli* Display and Single-Step Colony Assay

**Mieko Kato [1] and Yoshiro Hanyu [2],***

[1] Department of Biochemistry, Bio-Peak Co., Ltd., 584-70 Shimonojo, Takasaki 370-0854, Japan; mieko.kato@bio-peak.com

[2] Biomaterials Research Group, Health and Medical Research Institute, National Institute of Advanced Industrial Science and Technology (AIST), 1-1-1 Higashi, Tsukuba 305-8566, Japan

*  Correspondence: y.hanyu@aist.go.jp; Tel.: +81-298-61-5542

**Abstract:** The use of single-domain camelid antibodies, termed VHHs or nanobodies, has found increasing application in diagnosis, pharmaceutical development, and research because of their superior properties, such as small size, elevated stability, high water solubility, and excellent affinity for the antigen. Antigen-specific VHHs are generated by screening VHH display libraries via bio-panning. However, the bio-panning step needs to be repeated multiple times, which is time-consuming and laborious. Here, we developed a simple and rapid screening method that combined *Escherichia coli* display and a single-step colony assay to successfully identify positive clones from a naïve VHH library. The library was constructed from peripheral blood mononuclear cells of alpaca, and VHHs were displayed on the surface of *E. coli* using the inverse autotransporter intimin. Libraries enriched by magnetic cell sorting were screened directly using a single-step colony assay. Colonies formed on the hydrophilic filter and antigen-coated membrane. The expression of VHHs was induced, and those bound to the antigen on the membrane were detected as positive clones. Screening and identification of positive clones required only two days, which saves considerable time and resources compared to existing protocols.

**Keywords:** *E. coli*; display; VHH; colony assay; screening; antibody; intimin

## 1. Introduction

Antibodies are indispensable in diagnostics, therapeutics, and research, as they recognize target molecules with high affinity and specificity [1]. Most antibodies comprise two heavy and two light chains, both of which contribute to antigen binding [2]. The light chain is composed of two immunoglobulin (Ig) domains, and the heavy chain is composed of four domains. The N-terminal Ig domains of the light chain (VL) and heavy chain (VH) function as antigen-binding sites. Antibodies recognize a wide variety of antigens because the VH and VL sequences are highly variable. Fragments containing these variable domains can be expressed as recombinant proteins and are used as tools for antigen binding [2–4]. This is the case of Fab fragments composed of four Ig domains, including VH and VL, or single-chain variable fragments (scFvs), whose VH and VL are joined by a linker [5]. These smaller fragments are used extensively in pharmacological and basic research because their smaller size offers advantages over conventional antibodies. They can be expressed in *Escherichia coli* and easily modified using molecular techniques. However, antigen binding requires proper folding and assembly of VH and VL via hydrophobic interfaces, which is often difficult to achieve [6,7].

In addition to conventional antibodies, sharks and camelids possess unusual antibodies composed of only heavy chains [8]. The antigen-binding sites of these heavy-chain antibodies comprise only one variable Ig domain, and, therefore, they are termed VHHs, nanobodies, or single-domain antibodies. VHHs are advantageous because of their small size, elevated thermal stability, high water solubility, excellent affinity for antigens, high

expression, and good tissue penetration in vivo [9–11]. In most cases, good yields of VHHs have been obtained in *E. coli*, further promoting their use in biotechnological and pharmaceutical applications [12–14].

Antigen-specific VHHs can be selected from immune libraries with display technologies. Although different display methods are available (phage display, ribosomal display, yeast display, and *E. coli* display) [15,16], phage display remains the most used as it is suitable for large libraries, such as those containing $10^{6~11}$ clones [17,18]. The first step in VHH generation is the immunization of alpacas (*Lama pacos*) or llamas (*Lama glama*) [10]. After the last immunization, blood is collected, and peripheral blood mononuclear cells (PBMC) are purified. Total RNA is extracted from PBMC and used as a template for first-strand cDNA synthesis. The VHH-encoding region is amplified by PCR and inserted into a phagemid vector. A library of $10^{8~9}$ variants is constructed by transforming *E. coli* with these phagemids. Three rounds of bio-panning (i.e., selection) are performed on solid-phase-coated antigens to enrich for antigen-specific phages. Clones expressing antigen-specific VHHs are selected via a phage enzyme-linked immunosorbent assay (phage-ELISA), and positive clones are sequenced. The entire selection process takes approximately three months: two for immunization and one to identify positive clones. Immunization can be avoided when naïve or synthetic libraries are employed. A naïve library is constructed by collecting blood from several non-immunized animals, followed by the same procedure used for immune library construction [19]. Immunization leads to the amplification and production of specific clones. This natural process of antibody selection does not occur in non-immunized animals, and thus, a larger number of clones is necessary for the screen to allow the detection of the rare positive clones present in naïve libraries. As a naïve library is larger ($>10^9$) than an immune library, frequent panning is required to sieve through a much lower rate of positive clones [10]. The total selection period for naïve libraries is approximately three weeks. In the case of a synthetic library, this period can be shortened to two weeks, although the low rate of positive clones still implies extensive panning.

Repetitive panning is not only time-consuming and laborious, but it also limits the effectiveness of enrichment. Rapidly proliferating clones irrelevant to antigen binding often expand during amplification, thereby interfering with enrichment for positive clones. False-positive clones are also common in phage display methods and become a problem when the number of positive clones in the library is low [20–22].

To overcome these issues, we developed a single-step *E. coli* colony assay for screening antibody libraries [23,24]. In this assay, *E. coli* colonies carrying the antibody fragment library were formed on a hydrophilic filter covering an antigen-coated membrane placed on an agar plate. Upon addition of an inducer, antibody fragments were secreted as soluble molecules and reacted with the antigens coated on the membrane. By detecting antibody fragments bound to the antigen, we identified positive clones. The proposed method for screening antibody libraries is simple and rapid, as it does not involve the transfer of filters harboring colonies for antibody fragment expression. Moreover, the single-step colony assay identifies positive clones within one day and without the risk of false positives because the selection and identification of positive clones occur simultaneously. The only drawback at present is the library size, which is limited to approximately $10^{5~6}$.

Fernandez et al. have developed an *E. coli* display system using the inverse autotransporter intimin [25,26]. Intimin is a member of a large family of virulence-related bacterial adhesins [27]. Intimin is similar to autotransporters but has an opposite topology, with the N-terminal signal peptide followed by a transmembrane β-barrel domain with an internal peptide segment that connects to a surface-exposed passenger C-terminal region. They also used magnetic cell sorting (MACS) of *E. coli* carrying biotin-labeled antigens to select antigen-specific VHHs. They found that two rounds of MACS selection were sufficient to attain significant enrichment of antigen-specific VHH clones with high affinity, allowing them to identify various monoclonal VHHs [28].

In this study, we developed a new method for screening VHH libraries by combining an *E. coli* display with a single-step colony assay. As a result, we successfully screened a

large library and identified VHHs that bound to rabbit IgG within two working days and without incurring false positives.

## 2. Materials and Methods

### 2.1. Vector Construction for Displaying VHHs on the Surface of E. coli

VHHs were displayed on the surface of *E. coli* by fusing them with the N-terminal inverse autotransporter (intimin) [25]. The N-terminus of VHHs was fused to the C-terminus of the D0 domain of intimin. A linker was inserted between the D0 domain and VHH. A His-tag was added to the C-terminus of the VHH, along with restriction sites (Figure 1A). The N-terminus intimin sequence harbored a signal peptide, a lysine motif, as well as β-barrel, D0, and D00 domains (Figure 1A). The N-terminus of intimin, restriction sites, His-tag, and stop codon were synthesized (Twist Bioscience, South San Francisco, CA, USA) and inserted in the rhamnose-inducible expression vector pD884 (Atum, Newark, CA, USA), thus yielding the *E. coli* display vector pRham-IHis. The resulting recombinant fusion protein contained a His-tag at the C-terminus and was localized in the outer membrane of *E. coli* (Figure 1B).

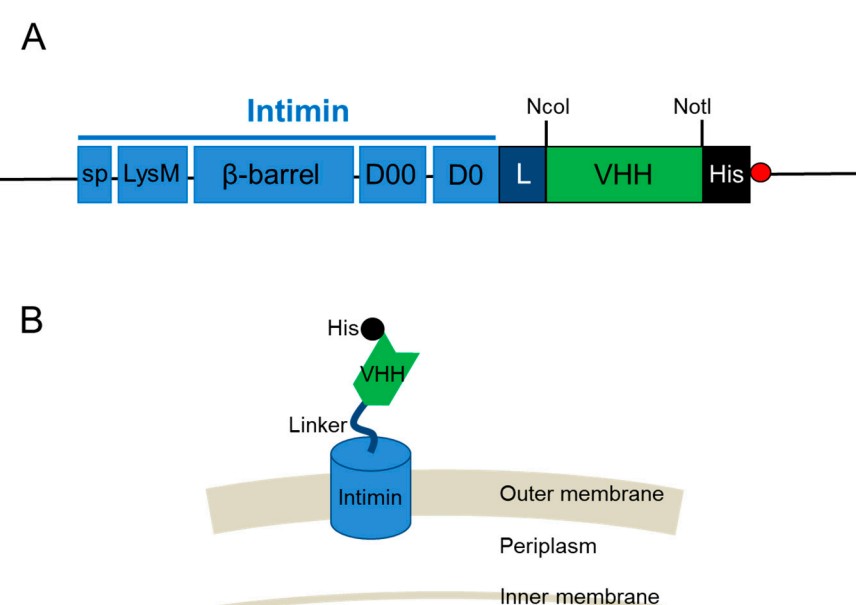

**Figure 1.** Construction of a fusion protein made of VHH and the β-barrel of the inverse autotransporter intimin. (**A**) Schematic representation of the fusion protein construct for surface display. sp, signal peptide; LysM, lysine motif; β-barrel, the β-barrel domain in intimin; D0 and D00, bacterial immunoglobulin-like domains in intimin; L, linker (G4S); VHH, VHH antibody; His, His-tag; red circle, stop codon. (**B**) Model of the fusion protein for *E. coli* surface display.

### 2.2. Library Construction

Briefly, $1 \times 10^9$ PBMC were isolated on a Lympho Spin (pluriSelect, Leipzig, Germany) from 300 mL blood of non-immunized alpacas. Total RNA was extracted using the RNeasy kit (Qiagen, Hilden, Germany). First-strand cDNA was synthetized from 1.0 mg of RNA using the Transcriptor reverse transcriptase (Roche Diagnostics, Indianapolis, IN, USA) and was then used as a template for the first PCR with the following primers: 5′-GTCCTGGCTGCTCTTCTACAAGG-3′ and 5′-GGTACGTGCTGTTGAACTGTTCC-3′ [29]. The PCR was performed with Q5 High-Fidelity DNA Polymerase (New England Bio-Labs, Ipswich, MA, USA) as follows: initial denaturation at 95 °C for 2 min; 25 cycles of 98 °C for 10 s, 57 °C for 30 s, and 72 °C for 1 min; and a final elongation step at 72 °C for 4 min. The PCR products were loaded onto a 1.5% agarose gel to separate conventional VH (~850 bp) from VHH (~600 bp). The latter were purified using a Nu-cleoSpin Gel and PCR Clean-up kit (Macherey Nagel, Düren, Germany). The purified

VHH sequences were amplified by PCR using primers that introduced the following NcoI and NotI restriction sites: 5′-GACTCCATGGATGGCTCAGGTGCAGCTGGTGGA-3′ and 5′-GACTGCGGCCGCTGAGGAGACGGTGACCTGGGT-3′. PCR conditions were as described above. PCR products (~400 bp) corresponding to the VHH fragments were identified on an agarose gel and purified with the NucleoSpin Gel and PCR Clean-up kit (Macherey Nagel, Düren, Germany). The amplified VHH fragments were digested with the NcoI and NotI restriction enzymes, ligated into the corresponding NcoI and NotI sites on the pRham-IHis vector, and purified using the NucleoSpin Gel and PCR Clean-up kit. Finally, the vectors were electroporated into NEB 10-beta Electrocompetent *E. coli* (New England BioLabs).

### 2.3. VHH Surface Display in E. coli

*E. coli* that displayed the VHH libraries were grown at 30 °C in 2 × YT liquid medium or on 2 × YT plates containing 1.5% (*w/v*) agar and supplemented with 0.1 mg/mL of carbenicillin and 2% (*w/v*) glucose. To induce the VHH fusion proteins, *E. coli* were grown on liquid 2 × YT-carbenicillin (without glucose) containing 5 mM of rhamnose.

### 2.4. MACS Selection of E. coli Displaying Antigen-Specific VHHs

Briefly, $5 \times 10^9$ colony-forming units (CFU) of *E. coli* were harvested by centrifugation at $4000 \times g$ for 5 min from an induced culture. Harvested *E. coli* were washed 2 times with 5 mL of phosphate-buffered saline (PBS, pH 7.0) at room temperature and resuspended in PBS supplemented with 0.5% (*w/v*) bovine serum albumin (BSA). *E. coli* at ~$2 \times 10^9$ CFU in a total buffer volume of 400 µL were mixed with 50 nM of biotinylated rabbit IgG (011-060-003, Jackson ImmunoReserach, West Grove, PA, USA). After incubation for 1 h at room temperature, *E. coli* were washed 2 times with 5 mL PBS and resuspended in 180 µL of PBS-BSA plus 20 µL of Anti-Biotin Magnetic Microbeads UltraPure (Miltenyi Biotec, Bergisch Gladbach, Germany). After 20 min of incubation at 4 °C, *E. coli* were again washed 2 times in PBS-BSA, resuspended in 1 mL of PBS-BSA, and loaded onto an LS column (Miltenyi Biotec) placed in a MidiMACS magnetic holder (Miltenyi Biotec). The columns were washed three times with 3 mL of PBS-BSA to remove unbound *E. coli*. The MACS column was then removed from the magnetic holder, and column-bound *E. coli* were eluted with 5 mL of PBS. Serial dilutions of the eluted fractions were plated on 2 × YT agar plates for CFU counting. For the subsequent selection cycle, the eluted fractions were loaded onto an MS column in a MiniMACS magnetic holder (Miltenyi Biotec). The column was washed 3 times with 1 mL of PBS-BSA to remove unbound *E. coli*, detached from the magnetic holder, and column-bound *E. coli* were eluted with 1 mL of PBS. Serial dilutions of the eluted fractions were plated on 2 × YT agar plates for CFU counting. After the second MACS cycle, bound *E. coli* were further screened using a single-step colony assay.

### 2.5. Screening of Positive Clones with a Single-Step Colony Assay

Selected clones with MACS were further screened with a single-step colony assay [23]. A nitrocellulose membrane (GE HealthCare, Chicago, IL, USA) was coated with the rabbit IgG (011-000-003, Jackson ImmunoReserach), blocked in PBS containing 5% nonfat dry milk, washed, and placed on a 2 × YT plate containing 0.1 mg/mL of carbenicillin. Then, Durapore filter (Merck Millipore, Burlington, MA, USA) was placed on the membrane. *E. coli* eluted from the MiniMACS column were suspended in 2 × YT medium, spread on the filter, and incubated at 30 °C for 16 h. After colonies had formed, 400 µL of 5 mM rhamnose (Merck, Rahway, NJ, USA) was sprayed. The plate was incubated at 30 °C for 6 h for the expression of VHH fusion protein. The filter was removed, placed in a fresh 2 × YT plate containing 0.1 mg/mL of carbenicillin and 1% glucose, and stored at 4 °C. The membrane was washed twice in PBS containing 0.05% Tween-20 (PBS-T), incubated for 2 h with horseradish peroxidase (HRP)-labeled anti-His antibody (Wako, Richmond, VI, USA) in PBS-T, and washed twice in PBS-T. After development with Immobilon Western Chemiluminescent HRP Substrate (Merck Millipore), the chemiluminescence signal was

detected and analyzed using a Chemi-Stage CC16mini imaging system (KURABO, Osaka, Japan). The filter with the colonies and the image of the chemiluminescence signal were superimposed, and positive colonies corresponding to the chemiluminescence signals were identified. Plasmids were purified from these positive clones using a NucleoSpin Plasmid EasyPure (Macherey-Nagel, Düren, Germany). The VHH sequences were determined using an ABI PerkinElmer 373A automated DNA sequencer (Applied Biosystems, Foster City, CA, USA).

### 2.6. Antigen Binding Evaluation by ELISA

Antigen binding of VHHs was evaluated with ELISA [23]. The clones were cultured at 30 °C in 2 × YT medium containing 0.1 mg/mL of carbenicillin. At an $OD_{600}$ of 0.6, the expression was induced by adding 5 mM of rhamnose. They were incubated at 30 °C for 6 h and then centrifuged at $4000\times g$ and 4 °C for 5 min. The pellets were collected, extracted with CelLytic (Merck), and analyzed by ELISA. Each well of Immuno MaxiSorp 96-well plates (Thermo Fisher Scientific, Waltham, MA, USA) was coated with 100 µL of 10 µg/mL rabbit IgG, blocked with 1% BSA for 2 h, washed with PBS-T, and filled with 100 µL of the extracted solution. The wells were washed three times with PBS-T. HRP-labeled anti-His antibody (Wako) was added to each well and incubated at room temperature for 1 h. The bound VHHs were detected by developing with SIGMAFAST OPD (Merck). The plates were read using the Model 680 microplate reader (Bio-Rad Laboratories, Hercules, CA, USA) at 450 nm. All experiments were carried out twice, and average signal intensity was used for analysis.

## 3. Results

### 3.1. Development of a Combination Screening

The VHH gene segments were amplified from PBMC and cloned into the pRham-IHis vector to create approximately $10^9$ independent clones (Figure 2). Upon induction with rhamnose, VHHs were displayed on the outer membrane of *E. coli* tied to the inverse autotransporter intimin.

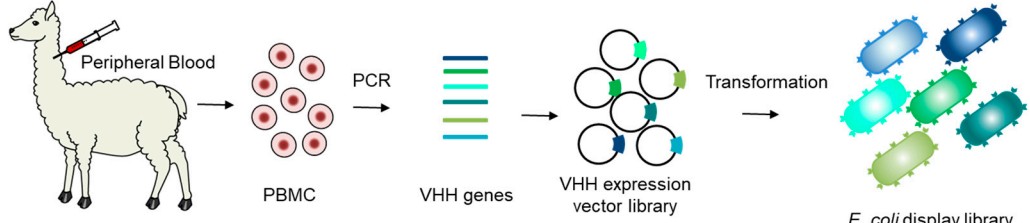

**Figure 2.** Steps required to generate an *E. coli* display library for VHH. PBMC was prepared from peripheral blood. Total RNA from PBMC was used to synthesize cDNA. VHH gene segments were amplified from cDNA and cloned into pRham-IHis, which was then transformed into *E. coli*.

The combination screening procedure based on the VHH display library is summarized in Figure 3. To select antigen-specific VHHs, MACS of *E. coli* incubated with antigens (biotin-labeled rabbit IgG) was performed as outlined in Figure 3A. *E. coli* cells were incubated with the biotinylated antigen, washed with PBS to remove the unbound antigen, and then incubated with anti-biotin magnetic microbeads. Next, the mixture was passed through an LS column held in a magnetic holder, which retained *E. coli* with a biotinylated antigen and anti-biotin microbeads but not *E. coli* with no antigen. The bound *E. coli* were eluted with PBS upon removal from the magnet and then passed through a small ferromagnetic column prior to single-step colony assay screening. The single-step colony assay is outlined in Figure 3B. The hydrophilic filter and antigen-coated nitrocellulose membrane were placed on an agar plate, and the VHH library eluted from the MACS column was spread directly onto the filter. VHH expression was induced by applying rhamnose. The colonies produced soluble periplasmic VHH fusions, and those showing affinity for the

antigen were captured by the antigen immobilized on the membrane. The upper filter carrying the viable colonies was transferred to a fresh agar plate and stored for the recovery of positive clones, whereas the lower membrane harboring the captured antibody fragments was developed to detect the antigen-binding activity by chemiluminescence. The colonies corresponding to the positive signals were then picked and cultured in a medium to isolate the plasmids encoding VHHs with affinity for the antigen.

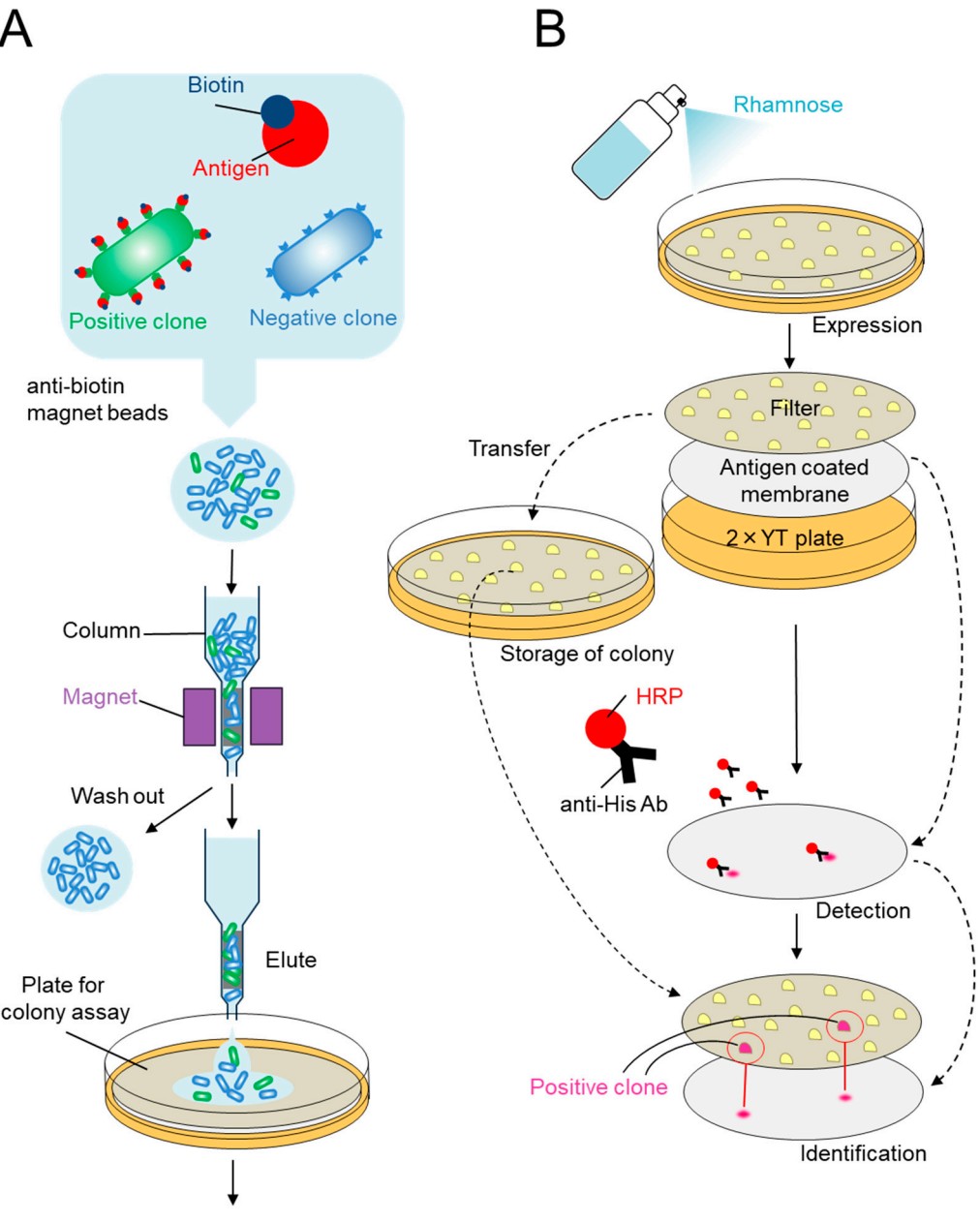

**Figure 3.** Schematic diagram of the procedure describing the combination of display and single-step colony assay. (**A**) Enrichment of positive clones from an *E. coli* display library using MACS. *E. coli* were incubated with the biotinylated antigen and anti-biotin magnetic beads. *E. coli* bound to the biotinylated antigen were captured in a MACS column held in a magnet, whereas those not bound to the antigen were washed out. Bound *E. coli* were eluted with PBS after removal of the column from

the magnet. CFU in the washed and bound fractions were determined by plating. (**B**) Single-step colony assay for the identification of positive clones. Intimin-VHH fusions were expressed. Part of expressed intimin-VHH fusions were diffused to the membrane. Those with the desired affinity bound the antigen beneath the colonies and were detected. Positive clones were identified as colonies matching the positive signals on the membrane.

### 3.2. Display of VHH on E. coli and Selection of Positive Clones

To identify the clones that bound to rabbit IgG from the naïve VHH library of alpaca, we employed a combination assay. A display library was constructed using $10^9$ PBMC from unimmunized alpacas. The expression of intimin-VHH fusion proteins was induced by rhamnose. Positive clones that bound the antigen were enriched by repeated MACS. The CFU of the input and eluent were measured: $2.0 \times 10^9$ *E. coli* were loaded onto the first column, and $1.1 \times 10^7$ were eluted. All eluted *E. coli* were loaded onto the second column, and $3.3 \times 10^4$ were eluted, thus resulting in a 60,000 times enrichment. To identify positive clones from this enriched library, a conventional antigen-binding ELISA would require more than 300 deep-well plate cultures. To overcome this hurdle, we screened the library using a single-step colony assay capable of handling a large number of clones.

### 3.3. Identification of Positive Clones Using the Single-Step Colony Assay

A single-step colony assay was performed to identify positive clones from the enriched display library (Figure 4). The selected $3 \times 10^4$ *E. coli* were spread on 10 plates ($3 \times 10^3$ bacteria per 10 cm diameter plate) and incubated for 16 h at 30 °C. Colonies varied in diameter from 0.5 to 0.9 mm. Next, rhamnose was sprayed on the upper filter to induce VHH expression. After incubation for 6 h at 30 °C, antigen binding of VHHs to the lower membrane was detected by chemiluminescence. Spots with high chemiluminescence intensity indicated binding of the expressed VHH fusions to the antigen. In the plate shown in Figure 4, there were 3 such spots (shown with arrows). The total number of colonies was 2764 on this plate. A total of 22 positive spots were identified on 10 plates. The percentage of positive colonies was approximately 0.067%. Ten of those with the strongest signal intensity were selected for further characterization.

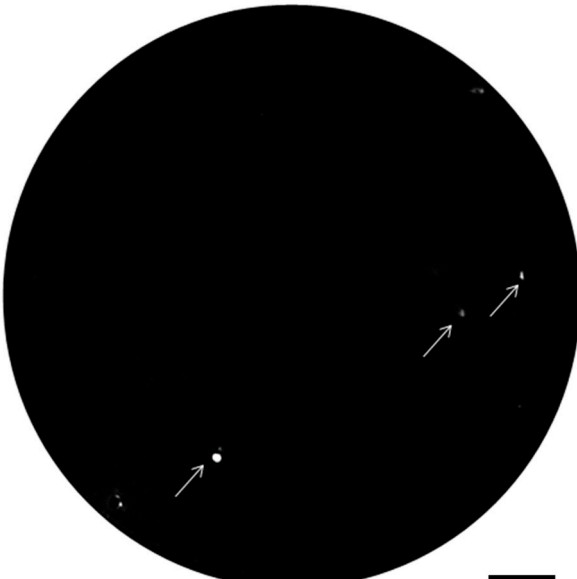

**Figure 4.** A representative nitrocellulose membrane showing clones from the VHH library positive for rabbit IgG. Antigen binding of VHHs to the membrane was detected by chemiluminescence. Three positive spots with the strongest signal intensities were selected for further experiments. Scale bar = 1 cm.

### 3.4. Antigen Binding of Positive Clones

Ten clones with the strongest signal and one with no signal were selected for further analysis. Each clone was identified by superimposing a filter onto the membrane chemiluminescent image. Cells were selected from the upper filter and cultured to express VHHs. The selected positive and negative clones were cultured at 30 °C in a 2 × YT medium containing 0.1 mg/mL of carbenicillin until they reached $OD_{600}$ = 0.6. Then, the cells were incubated at 30 °C for 6 h in the presence of 5 mM of rhamnose. Thereafter, they were centrifuged at 4000× *g* and 4 °C for 5 min, and the pellets were collected and lysed with CelLytic solution. The reactivity of the extract against rabbit IgG and BSA was examined by ELISA (Figure 5). Each positive clone exhibited antigen-specific binding activity, whereas no binding was detected in the negative clone or on an uncoated ELISA plate blocked with BSA. Thus, the clones expressing anti-rabbit IgG VHHs exhibited binding affinity and specificity for the antigen.

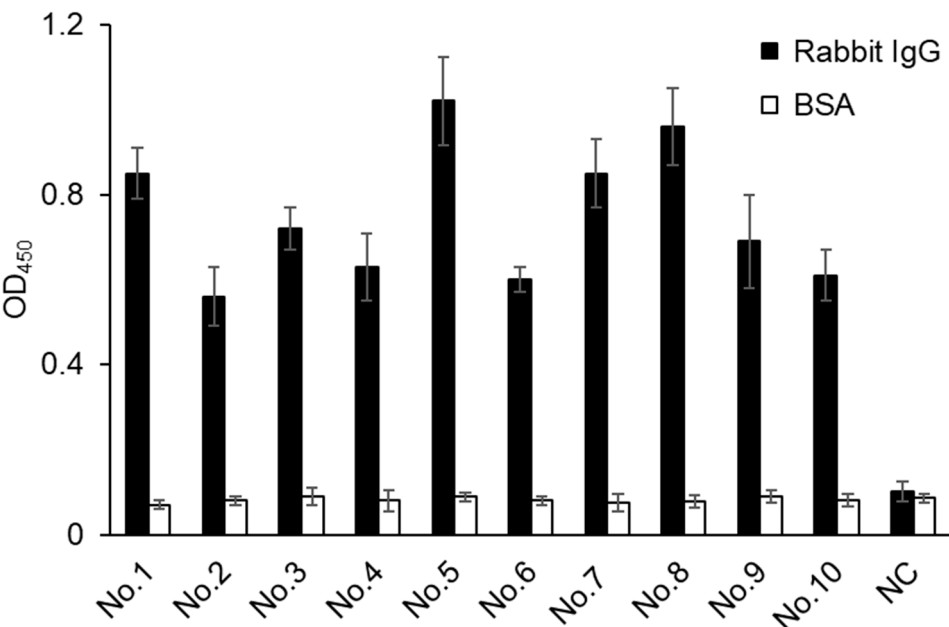

**Figure 5.** ELISA results showing the reactivity of culture lysate from 10 positive clones and 1 negative clone against rabbit IgG and BSA (control). The negative clone (NC) was picked at random from colonies that exhibited no chemiluminescence signal. Data represent the mean of three replicates; error bars represent the standard deviation.

### 3.5. Sequence Analysis of Positive Clones

The DNA of each VHH from the 10 clones (Figure 5) was sequenced. The inferred amino acid sequences are reported in Figure 6. Every clone contained a complete VHH structure consisting of three complementary-determining regions: CDR1, CDR2, and CDR3. All sequences were unique, indicating that the diversity of the VHH library was maintained throughout the assay. These results confirmed the successful isolation of genes encoding intimin-VHH fusions with binding affinity for the antigen.

```
                                            CDR1                              CDR2
No.1    1 QVQLVQAGGGSVQTGGSLRLSCAASGRTFGSLDMGWYRQAPGKAREFVARISRNGGSTRYADSVKGRFTI 70
No.2    1 .....ES...L..A.D..........|...ST...|.......RE.D...A.|TWSD...Y|.......... 70
No.3    1 .....ES...L..P........TV..|F..DNYW.Y|.V.LT...GL.W.SA.|.SG..G.S|.I........ 70
No.4    1 .....ES...L..A...G...T...|G..STY..A|........E.....S.|TWSAHR.|....FG...... 70
No.5    1 .....ES...L..A.D......E...|F..SNYA..|.F......E....SS.|T.T...KQF|.......... 70
No.6    1 .....ES...L..A...........|..S.Y...|.......E....G.|WS.D..YNG|........ 70
No.7    1 .....ES...L..A.D......S...|.L.N.YA.A|.F.....NE.....A.|.WIIRN.H|..P........ 70
No.8    1 .....ES......A.D......V...|L..SRNA..|.F..GL..E...L..AL|AW..AR.Y|.S........ 70
No.9    1 .....ES...L..A...........|.P...YG..|.F......E....T.|NWS....S|..S........ 70
No.10   1 .....ES...L..P.........E...|F..STYY.S|.V..V...GL.W....|K.ETDT.Y|.......... 70

                                            CDR3
No.1   71 SRDGANNTVFLQMNSLKPEDTAIYTCNADDPYYGGIH---------WGRGTQVTVSS           118
No.2   71 ...N.K...Y...............Y...|VPFSQ.PDY---------|..Q........          118
No.3   71 ...N.KD.LY..........A.L.Y.A-|RSSGRWQKKGYDY-----|..Q........          121
No.4   71 ...N.K.............R.......V.K.|V.EHSR.VSSDY------|..Q........          121
No.5   71 ...N.K...Y...........T.Y.A.|KLYFGFDLSTLSHYDY--|..Q........          125
No.6   71 ...N.K...Y................Y.RP|RYSSSWYGEGADFGS---|..Q........          124
No.7   71 T..N.KEM.Y...........V.Y.A.|TFNEP.DSDVPGWYDS--|..Q........          125
No.8   71 ...N.E...Y.......S....V.Y.A.|TRDAELTVVLPHGPFDYR-|.Q........          126
No.9   71 ...N.K...N...........V.Y.A.|RGYAGASNLPREYSY---|..Q........          124
No.10  71 ...N.KK.LY............L.Y.A-|R..SDAAWAYEY------|..Q........          120
```

**Figure 6.** Amino acid sequence alignments of the selected clones. The complementary-determining region CDR1, CDR2, and CDR3 sequences are highlighted. Dots and dashes indicate identical amino acids and deletions, respectively.

## 4. Discussion

VHHs are a novel and unique class of antigen-binding fragments derived from naturally occurring heavy-chain-only antibodies present in camelids. VHHs have significant advantages over conventional antibodies, such as small size, good stability, high solubility, and excellent affinity for the antigen [9]. Antigen-specific VHHs are usually screened and selected from libraries using display technologies such as phage display, which can handle large libraries [18]. However, the selection of positive clones by panning is time-consuming and laborious. In the case of naïve and synthetic libraries, the probability of positive clones is also extremely low, thereby requiring extensive panning. This process can be further hindered by the presence of clones with fast growth but without affinity for the antigen [20–22]. Alternatively, one could start from an extremely large number of colonies, culture them, induce the expression of VHH, and examine antigen binding using phage-ELISA.

In this study, we developed a simple and rapid screening method for large libraries, such as naïve libraries. The protocol allows for efficient identification of possibly very rare positive clones by combining *E. coli* display and a single-step colony assay. First, we screened a large library of $2 \times 10^9$ *E. coli*, which we enriched to $3.3 \times 10^4$ *E. coli*, and from it, we identified 22 positive clones within two days and without any false positives. Because we chose the *E. coli* display method instead of the phage display, we screened the VHH library directly using a single-step colony assay without having to switch a vector or host. With phage display, infection of *E. coli* by selected phages and the production of phages for the next panning by helper phage infection are required, thereby lengthening the procedure. *E. coli* display can be made even more efficient by selecting via MACS [27]. Here, we used a rhamnose-inducible vector; however, the screening process could be shortened by introducing auto-induction in future [30].

Fernandez et al. developed an *E. coli* display and screening system using the inverse autotransporter intimin [25,26], which they employed to successfully identify antigen-specific VHHs with high affinity for various antigens [27,31–33]. The same method was employed here for the combination screening; however, expression and selection were performed slightly differently. For expression, instead of an IPTG-inducible vector, we used a rhamnose-inducible vector (pRham-IHis), which diminished the risk of cell lysis. Overexpressed antibody fragments in *E. coli* are often toxic to the host and sometimes lead to cell lysis. The expression of VHHs with a rhamnose-inducible expression vector is

tightly tunable by varying the rhamnose concentration, thus allowing precise protein levels and avoiding the risk of overexpression [34–36]. In addition, the $G_4S$ linker was inserted between the D0 domain and VHH in pRham-IHis. To select positive clones, instead of plating the selected clones after the first column, collecting them the next day, and applying them to the second column; here, we applied the library on the MACS columns sequentially. The original method maintained library size after the first column and relied on fluorescent-activated cell sorting for positive clone identification following the MACS enrichment [27]. Such an approach was effective but time-consuming and dependent on expensive instrumentation. In our case, we did not need to enrich the positive clones as much as possible because we could identify them by a subsequent colony assay.

In our combination assay, the first screening (selection by *E. coli* display) was performed against the biotinylated antigen suspended in solution, whereas the second screening (identification by a single-step colony assay) was performed against the antigen bound to a membrane. Clones that recognized both types of antigens were identified. Positive clones identified by the combination screening did not include any false positives, such as rapidly growing clones without affinity to the antigen typically found during repeated panning [21].

This combination method is particularly useful when the ratio of positive clones in the library is low and a large library must be screened. After screening with a display system, ELISA can be used to assess the antigen binding of VHHs from an extremely large number of clones. For example, if the ratio of positive clones was $1 \times 10^{-8}$ in the original library of $1 \times 10^9$ *E. coli* and MACS enrichment of positive clones was 10,000 times, the ratio of positive clones among the selected libraries would be approximately $1 \times 10^{-4}$. In this case, $1 \times 10^4$ clones would have to be examined for antigen binding by ELISA to identify a single positive clone and $1 \times 10^5$ for 10 positive clones. Overall, 100 deep-well plate cultures would be required for each positive clone. Instead, our single-step colony assay was able to select 10 positive clones from only 20 culture plates.

Combination screening is a powerful tool for screening naïve libraries, whose ratio of positive clones is very low. However, positive clones from naïve libraries have dissociation constants in the low micromolar range, which is insufficient for most antibody applications [37]. When the affinity and specificity of the identified VHHs are insufficient, they can be improved by molecular evolution following the selection procedure [38]. We did not quantitatively estimate affinity, as it fell outside the scope of the present study. However, the affinity constant could be measured by SPR in future investigations.

In the single-step colony assay, we just measure the signal intensities of spots and select the clones with strong signals. The signal intensity of the spot is determined by integrating the signal of the whole spot. The size and peak intensity of the spot are important, respectively. We think that the size might correspond to the amount of expression, and the peak intensity might correspond to the affinity. The amount of expression of best clones in terms of affinity is not necessarily the highest but often low. By selecting the peak intensity and spot area, we might obtain various positive clones regarding affinity and productivity.

In the combination assay, neither the vector nor the host underwent any changes. In the single-step colony assay, antibody fragments containing the signal peptide were expressed and transported to the periplasm [23]. Some periplasmic antibody fragments were released from the periplasm and bound to the antigen on the membrane. They were successfully detected using the HRP-labeled anti-His antibody [39]. Here, intimin-VHH fusion proteins were expressed, transported to the periplasm, and displayed on the outer membrane in a single-step colony assay. To bind antigens to the nitrocellulose membrane under the filter, intimin-VHH fusion proteins must be secreted via either leakage [40–42], cell lysis [43,44], outer membrane vesicle formation [45,46], or proteolytic cleavage [47]. At present, the exact secretion mechanism remains unknown. The assay using *E. coli* strains without the proteolytic enzyme [47] or the hypervesiculating *E. coli* that forms more outer membrane vesicles [45] as the host might provide clues regarding the elucidation of mechanisms.

We have applied the combination screening to the VHH naïve library. Valuable monoclonal scFv, whose affinity is often higher than that of VHHs [6], can be obtained by screening the scFv display library. However, it is not applicable to scFvs since we could not display scFvs efficiently. While some scFv clones have been successfully displayed on *E. coli* [48–51], efficient display of scFvs remains challenging [52]. To overcome this hurdle, it is necessary to identify the most suitable outer membrane fusion partners for scFvs display on *E. coli*. This will help identify positive clones by combination screening, as the scFv library has already been successfully screened with the single-step colony assay.

**Author Contributions:** Conceptualization, Y.H.; validation, Y.H. and M.K.; formal analysis, M.K.; investigation, M.K.; data curation, Y.H. and M.K.; writing—original draft preparation, Y.H.; writing—review and editing, Y.H.; project administration, Y.H. All authors have read and agreed to the published version of the manuscript.

**Funding:** This research received no external funding.

**Data Availability Statement:** Data are contained within the article.

**Conflicts of Interest:** Mieko Kato is an employee of Bio-Peak Co., Ltd., Author Mieko Kato has received research grants from Bio-Peak Co., Ltd.

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
