# Peer review of "Combination Screening of a Naïve Antibody Library Using E. coli Display and Single-Step Colony Assay"

_2673-8007, doi:10.3390/applmicrobiol4010016_

Round 1
Reviewer 1 Report
Comments and Suggestions for Authors
Kato and Hanyu describe the selection of single-domain camelid antibodies binding to rabbit IgG with a two-step method, termed “combination screening”. The first step (Escherichia coli display) has been already published by others and the second step (single-step colony assay) is reminiscing to what we have done about 30 years ago for the screening of lambda phage librairies encoding Fab fragments. The originality of the presented work is the combination of both that allows apparently faster selection of clones embedded in naïve libraries. I have no major criticism but I feel that the following points should be considered to render this manuscript even more convincing.
1. The intinimin protein. This protein which was used to display the VHHs at the outer bacterial membrane is clearly of importance for the MACS selection, but for the second approach it is unclear how the VHHs when bound to the cells can diffuse through the filter for binding to the antigen coated on the nitrocellulose membrane. On one hand, you would need immobilised VHHs and on the other soluble ones. I would like to see a comment on this and also a comment on the “natural” role and behavior of intinimin in bacteria. What is a autotransporter protein? In contrast to that shown in Figure 1B, the authors write “Intinimin-VHH fusions were expressed and secreted by E. Coli” (line 242). Could this a bit clarified?
2. Figure 4. The nitrocellulose membrane shows several spots: 3 indicated with an arrow and two others of low intensity at the edge of the filter (not indicated with an arrow). Since the authors have used only those with the strongest signal throughout the study, it might be interesting to test with a second colony assay if the ones giving a low signal (middle right of the filter in Figure 4) correspond to true positive clones or not. This clarification would certainly be a plus to claim that the method described is also useful to the selection of lowly-expressed VHHs from a naïve library. And also, we often found that the “best” clones in terms of affinity (generally determined at the end of the study) correspond in fact to lowly-expressed VHHs. On the other hand, those showing strong signals may have a low affinity, as their binding capacity is somewhat compensated by the amount of molecules that are secreted.
3. When reading the Discussion section, I was confused when reading the last paragraph of this section (lines 377-383). “ We identified a positive clone…”, whereas 10 positive clones are shown in Figure 5! “By treating the scFv and Fab library…..”, this is not the case here. The whole paragraph should be seriously revised or even deleted.
Minor point (lines 66 to 68): immunisation leads to the amplification and production of specific clones. This natural process of antibody selection is not going on in non-immunised animals and thus a larger number of clones are necessary for the screen to allow the detection of the rare positive clones present in naïve libraries. Please, rewrite the text to clarify.
Author Response
We are grateful to reviewer for the critical comments and useful suggestions that have helped us improve our paper considerably. As indicated in the responses that follow, we have taken all comments and suggestions into account in the revised version of our paper.
Reviewer: 1
Comments to the Author:
- The intinimin protein. This protein which was used to display the VHHs at the outer bacterial membrane is clearly of importance for the MACS selection, but for the second approach it is unclear how the VHHs when bound to the cells can diffuse through the filter for binding to the antigen coated on the nitrocellulose membrane. On one hand, you would need immobilised VHHs and on the other soluble ones. I would like to see a comment on this and also a comment on the “natural” role and behavior of intinimin in bacteria. What is a autotransporter protein? In contrast to that shown in Figure 1B, the authors write “Intinimin-VHH fusions were expressed and secreted by E. Coli” (line 242). Could this a bit clarified?
Author response and action taken:
This is an excellent suggestion. This issue is very important in the single-step colony assay. The antibody fragments with the periplasmic signal peptides are expressed and transported to the periplasm in the single-step colony assay. A part of periplasmic antibody fragments is released to the outside from the periplasm. These released antibody fragments are bound to the antigen on the membrane and successfully detected by the HRP-labeled anti-His antibody. We don’t know how the antibody fragments diffuse from the periplasm. However, the existence of antibody fragments on the membrane was confirmed by detecting the antibody fragments on the membrane. When we express antibody fragments with the same vector of the single-step colony assay in liquid culture, we also found the antibody fragments in the culture media. They might be secreted via either leakage or cell lysis. Part of expressed proteins were detected in the culture media when they are expressed in the periplasm.
When we screen E. coli display library with the single-step colony assay, we found soluble fraction of Intimin-VHH fusions. We also used the periplasmic signal sequence for E. coli display. Intimin-VHH fusions are transported to the periplasm and then incorporated in the outer membrane. Intimin-VHH fusions are detected not only on the surface of E. coli but also in the culture media in case of the liquid culture and on the membrane in case of the single-step colony assay. Intimin-VHH fusions might be secreted like the case of periplasmic expression by leakage or cell lysis. Or VHH are cleaved from the Intimin-VHH fusions on the outer membrane by proteolytic enzyme. On the other hand, the outer membrane vesicle formation was observed. Intimin-VHH fusions on the outer membrane vesicle might bind to the antigen on the membrane since the diameter of the outer membrane vesicle is smaller than the pore of filter that we use with single-step colony assay. We should clarify this mechanism in the next study. We have described these situations in Discussion (line 388). We think that we should discuss these points more. We have added the following sentences in the discussion part (line 388).
To bind antigens on the nitrocellulose membrane under the filter, intimin-VHH fusion proteins must be secreted via either leakage [40–42], cell lysis [43,44], outer membrane vesicle formation [45,46] or proteolytic cleavage [47]. At present, the exact secretion mechanism remains unknown. The assay with using E. coli strains without the proteolytic enzyme [47] or the hypervesiculating E. coli that forms more outer membrane vesicles [45] as the host might provide the clues for elucidation of mechanisms.
Figure caption of Figure 1B was changed as follows (line 246).
Before revision
Intinimin-VHH fusions were expressed and secreted by E. Coli.
After revision
Intimin-VHH fusions were expressed. Part of expressed intimin-VHH fusions were diffused to the membrane.
We have added the following sentences in the introduction part in order to explain Intimin itself (line92).
Intimin is a member of a large family of virulence-related bacterial adhesins [27]. Intimin is similar to autotransporters but has an opposite topology, with the N-terminal signal peptide followed by a transmembrane b-barrel domain with an internal peptide segment that connects to a surface-exposed passenger C-terminal region.
Comments to the Author:
- Figure 4. The nitrocellulose membrane shows several spots: 3 indicated with an arrow and two others of low intensity at the edge of the filter (not indicated with an arrow). Since the authors have used only those with the strongest signal throughout the study, it might be interesting to test with a second colony assay if the ones giving a low signal (middle right of the filter in Figure 4) correspond to true positive clones or not. This clarification would certainly be a plus to claim that the method described is also useful to the selection of lowly-expressed VHHs from a naïve library. And also, we often found that the “best” clones in terms of affinity (generally determined at the end of the study) correspond in fact to lowly-expressed VHHs. On the other hand, those showing strong signals may have a low affinity, as their binding capacity is somewhat compensated by the amount of molecules that are secreted.
Author response and action taken:
This is an excellent suggestion. This issue is very important in the single-step colony assay. The signal intensity of spot is determined by integrating the signal of whole spot. The size and peak intensity of spot are also important. We think that the size might be corresponding to the amount of expression and the peak intensity might be corresponding to the affinity. As the reviewer pointed out, the expression of best clones in terms of affinity is often low. We just measure the signal intensities of spots and select the clones with the strong signals now. We think that we could not select the clones with the highest affinity by selecting the clones based on only the signal intensity. It would be very important to analyse data from single-step colony assay and select the clones based on these analysed data. We have added the following sentences in the discussion part (line 374).
In the single-step colony assay, we just measure the signal intensities of spots and select the clones with the strong signals. The signal intensity of spot is determined by integrating the signal of whole spot. The size and peak intensity of spot are important, respectively. We think that the size might be corresponding to the amount of expression and the peak intensity might be corresponding to the affinity. The amount of expression of best clones in terms of affinity is not the highest necessarily but often low. By selecting the peak intensity and spot area, we might obtain variety of positive clones regarding affinity and productivity.
Comments to the Author:
- When reading the Discussion section, I was confused when reading the last paragraph of this section (lines 377-383). “ We identified a positive clone…”, whereas 10 positive clones are shown in Figure 5! “By treating the scFv and Fab library…..”, this is not the case here. The whole paragraph should be seriously revised or even deleted.
Author response and action taken:
This is an excellent suggestion. We have revised this part extensively (line 395).
Before revision
We successfully identified a positive VHH clone in a naïve library. By treating the scFv or Fab library, valuable monoclonal antibody fragments, whose affinity is often higher than that of VHHs, can be obtained [6]. While some clones have been successfully displayed on E. coli [47–50], efficient display of scFvs remains challenging [51].
After revision
We have applied the combination screening to VHH naïve library. It is not applicable to scFvs since we could not display scFvs efficiently. However, valuable monoclonal scFv, whose affinity is often higher than that of VHHs, can be obtained [6] by screening the scFv library. While some scFv clones have been successfully displayed on E. coli [48–51], efficient display of scFvs remains challenging [52].
Comments to the Author:
Minor point (lines 66 to 68): immunisation leads to the amplification and production of specific clones. This natural process of antibody selection is not going on in non-immunised animals and thus a larger number of clones are necessary for the screen to allow the detection of the rare positive clones present in naïve libraries. Please, rewrite the text to clarify.
Author response and action taken:
This is an excellent suggestion. We have added the following sentences in the introduction part (line 66).
Before revision
However, as a naïve library is larger (>109) than an immune library, frequent panning is required to sieve through a much lower rate of positive clones [10].
After revision
Immunisation leads to the amplification and production of specific clones. This natural process of antibody selection is not going on in non-immunised animals and thus a larger number of clones are necessary for the screen to allow the detection of the rare positive clones present in naïve libraries. As a naïve library is larger (>109) than an immune library, frequent panning is required to sieve through a much lower rate of positive clones [10].
Reviewer 2 Report
Comments and Suggestions for Authors
* Although the term "antigen" is mentioned many times, it is often unclear, what in fact was the antigen. Is it correct, that the antigen was rabbit IgG? This should be repeated also in the respective chapter titles. In addition, the sample of "rabbit IgG" should be specified in more detail, at least with an order number.
In Fig. 4 it would be interesting to know the total number of clones on this area. Perhaps a photograph is available of the membrane.
Fig. 5: The paper would gain a lot, if the clones would be characterized with their affinity constants, e.g. by SPR.
Author Response
We are grateful to reviewer for the critical comments and useful suggestions that have helped us improve our paper considerably. As indicated in the responses that follow, we have taken all comments and suggestions into account in the revised version of our paper.
Reviewer: 2
Comments to the Author:
* Although the term "antigen" is mentioned many times, it is often unclear, what in fact was the antigen. Is it correct, that the antigen was rabbit IgG? This should be repeated also in the respective chapter titles. In addition, the sample of "rabbit IgG" should be specified in more detail, at least with an order number.
Author response and action taken:
We thank the reviewer for the valuable comment. We used the rabbit IgG as an antigen. We have changed the following sentences in the introduction part (line 100).
Before revision
In this study, we developed a new method for screening antibody naïve libraries by combining an E. coli display with a single-step colony assay. As a result, we successfully screened a large library and identified positive clones within two working days and without incurring in false positives.
After revision
In this study, we developed a new method for screening VHH naïve libraries by combining an E. coli display with a single-step colony assay. As a result, we successfully screened a large library and identified VHHs that bound to rabbit IgG within two working days and without incurring in false positives.
We also specified the rabbit IgG and biotinylated rabbit IgG by adding the production numbers in the material and methods section (line 153 and 171).
rabbit IgG (011-000-003, Jackson ImmunoReserach)
biotinylated rabbit IgG (011-060-003, Jackson ImmunoReserach)
Comments to the Author:
1, In Fig. 4 it would be interesting to know the total number of clones on this area. Perhaps a photograph is available of the membrane.
Author response and action taken:
We thank the reviewer for the valuable comment. We have added the following sentences in the result part (line 267).
Before revision
Spots with high chemiluminescence intensity (see arrows in Figure 4) indicated binding of the expressed VHH fusions to the antigen.
After revision
Spots with high chemiluminescence intensity indicated binding of the expressed VHH fusions to the antigen. In the plate shown in Fig. 4, there were 3 such spots (shown with arrows). The total number of colonies was 2764 in this plate.
Comments to the Author:
2, Fig. 5: The paper would gain a lot, if the clones would be characterized with their affinity constants, e.g. by SPR.
Author response and action taken:
We thank the reviewer for the valuable comment. We have changed the following sentences in the discussion part (line 373).
Before revision
However, it could be measured in future investigations.
After revision
However, affinity constant could be measured by SPR in future investigations.